# Enhancing Oil Recovery in Low-Permeability Reservoirs Using a Low-Molecular Weight Amphiphilic Polymer

**DOI:** 10.3390/polym16081036

**Published:** 2024-04-10

**Authors:** Yang Yang, Youqi Wang, Yiheng Liu, Ping Liu

**Affiliations:** 1State Key Laboratory of Shale Oil and Gas Enrichment Mechanisms and Effective Development, Beijing 100083, China; 2Research and Development Center for the Sustainable Development of Continental Sandstone Mature Oilfield by National Energy Administration, Beijing 100083, China; 3College of Energy, Chengdu University of Technology, Chengdu 610059, China; liuyiheng2666@163.com

**Keywords:** enhanced oil recovery, low molecular weight, amphiphilic polymer, interfacial activity, low-permeability reservoirs

## Abstract

Polymer flooding has achieved considerable success in medium–high permeability reservoirs. However, when it comes to low-permeability reservoirs, polymer flooding suffers from poor injectivity due to the large molecular size of the commonly used high-molecular-weight (high-MW) partially hydrolyzed polyacrylamides (HPAM). Herein, an amphiphilic polymer (LMWAP) with a low MW (3.9 × 10^6^ g/mol) was synthesized by introducing an amphiphilic monomer (Allyl-OP-10) and a chain transfer agent into the polymerization reaction. Despite the low MW, LMWAP exhibited better thickening capability in brine than its counterparts HPAM-1800 (MW = 1.8 × 10^7^ g/mol) and HPAM-800 (MW = 8 × 10^6^ g/mol) due to the intermolecular hydrophobic association. LMWAP also exhibited more significant shear-thinning behavior and stronger elasticity than the two counterparts. Furthermore, LMWAP possesses favorable oil–water interfacial activity due to its amphiphilicity. The oil–water interfacial tension (IFT) could be reduced to 0.88 mN/m and oil-in-water (O/W) emulsions could be formed under the effect of LMWAP. In addition, the reversible hydrophobic association endows the molecular chains of LMWAP with dynamic association–disassociation transition ability. Therefore, despite the similar hydrodynamic sizes in brine, LMWAP exhibited favorable injectivity under low-permeability conditions, while the counterpart HPAM-1800 led to fatal plugging. Furthermore, LMWAP could enhance oil recovery up to 21.5%, while the counterpart HPAM-800 could only enhance oil recovery by up to 11.5%, which could be attributed to the favorable interfacial activity of LMWAP.

## 1. Introduction

The low-permeability reservoirs host more than two-thirds of the total hydrocarbon reserves in China and have become the strategic substitute for conventional energy sources [1]. However, low-permeability reservoirs suffer from poor porosity, low permeability, small laryngeal radius, and significant heterogeneity. As a result, the water displacement efficiency in low-permeability reservoirs is low and a large amount of residual oil cannot be mobilized, which leads to a low oil recovery [2,3,4,5,6,7,8]. To meet an increasing energy demand and to prevent the deterioration of the oil production rates from mature fields, the development of enhanced oil recovery (EOR) methods for low-permeability reservoirs has become highly important.

Polymer flooding has been proven to be an efficient EOR method to increase oil production and reduce the water cut by various mechanisms [9,10,11,12,13,14,15,16,17,18,19,20], including controlling the mobility ratio of water and oil, improving the viscoelasticity of water, and improving sweep efficiency. The polymer flooding method has been applied in many fields across the world, both onshore and offshore. Renouf [21] reported 32 cases of polymer flooding in western Canada, and incremental recovery ranged from 0.5 to 14% of the original oil in place over periods lasting between 1 and 9 years. Sixteen of the thirty-two polymer floods reduced their water–oil ratio (WOR), and screening criteria for polymer floods should be adjusted from the previous recommendations of recovering oils with viscosity less than 150 mPa·s and API gravity above 15°. Delamaide [22] summarized ten polymer flood projects in Canada, Argentina, China, Oman, and Suriname and found that the key screening criteria for polymer flooding are oil viscosity (lower than 5000 cp), low reservoir temperature (less than 80 °C preferred), no bottom aquifer, high permeability, low salinity and low hardness water, sandstone reservoirs, and mobile oil saturation (>30%). Sheng et al. [23] collected and surveyed 733 polymer-flooding projects in 24 countries worldwide. The results showed that the median incremental oil-recovery factor was 6.7% and the median decrease in water cut after polymer injection was 13%.

In the design of a polymer flooding pilot test, the selection of polymer is the most crucial problem to be resolved. Hydrolyzed polyacrylamide (HPAM) is the most commonly used polymer in polymer flooding. Normally, high-molecular-weight (MW) HPAM has a higher thickening capability and higher permeability-reducing factor than low MW HAPM. For the same amount of polymer injected, the HPAM with higher MW would result in a higher oil recovery. However, the MW of HPAM must be small enough for the HPAM molecules to enter and propagate effectively through the reservoir rock. For a given permeability and a pore throat size, a threshold MW exists, above which HPAM molecules exhibit poor injectivity. The MW of the HPAM used in polymer flooding is usually higher than 12 million daltons, which makes the use of HPAM applicable in reservoirs with an average permeability greater than 100 mD [24,25,26,27,28,29,30]. When it comes to low-permeability reservoirs with an average permeability lower than 50 mD, an HPAM with medium and high WM is not applicable. The use of low-WM HPAM, however, would weaken the thickening capability and mobility control ability of HPAM.

In order to achieve polymer flooding in low-permeability reservoirs, many functionalized polymers have been developed. Gong et al. [31] synthesized encapsulated polymers with different shell–core ratios and the encapsulated polymers exhibited delayed thickening capability. The encapsulated polymers with a shell–core ratio of 0.25 had a uniform particle size distribution (297.8~531.8 nm), and the viscosity of the fluid increased from 1 mPa∙s to 12.84 mPa∙s after 9 d. This feature helps to improve the injectability of the fluid into the low-permeability reservoir. Mejía et al. [32] studied the potential of polyethylene oxide (PEO) as an oil displacement agent for polymer flooding. The results demonstrated that high MW PEO solutions had favorable injectivity in low-permeability (~20 md) carbonate cores and improved oil recovery by 15% in an artificially fractured limestone core. Zhang et al. [33] proposed the use of a self-adaptive polymer (SAP) in low-permeability reservoirs to alleviate issues such as poor injectivity and ease of mechanical degradation for HPAM. SAP can be smoothly injected into the 60 mD cores and improve oil recovery up to 18.7%, which may open a new pathway for the molecular design of polymers used for EOR in low-permeability reservoirs. Despite extensive studies, up to now, the application of polymer flooding for EOR in low-permeability reservoirs is still very challenging.

In this study, an amphiphilic monomer Allyl-OP-10 was synthesized based on a commercial surfactant OP-10. Then, an amphiphilic polymer (LMWAP) with a low MW was synthesized by the copolymerization of acrylamide (AM), acrylic acid (AA), and Allyl-OP-10. LMWAP was characterized through FTIR, ^1^H-NMR, SEM, and static–dynamic light scattering. The thickening capability, rheology, and viscoelasticity of LMWAP were comparatively studied with two counterparts, HPAM-800 and HPAM-1800. The molecular weights of HPAM-800 and HPAM-1800 were 8 × 10^6^ g/mol and 18 × 10^6^ g/mol, respectively. Oil–water interfacial tension and emulsification tests were conducted to study the interfacial activity of LMWAP. The injectivity and oil displacement ability of LMWAP under low-permeability conditions were studied by core flooding experiments. The results of this study could provide some enlightenment on the molecular design of polymers used for EOR in low-permeability reservoirs.

## 2. Experimental Section

### 2.1. Materials

Acrylamide (AM), acrylic acid (AA), sodium hydroxide, ethanol, bromo-propylene, OP-10, sodium formate, and 2,2′-Azobis(2-methylpropionamidine) dihydrochloride (V50) were purchased from Aladdin Chemical Reagent Company (Shanghai, China). All the chemicals were used as received without further purification. High-MW HPAM (MW = 1.8 × 10^7^ g/mol) and medium-WM HPAM (MW = 8 × 10^6^ g/mol) were provided by Jiangsu Feymer Technology Co., Ltd. (Zhangjiagang, China). Crude oil and formation water were provided by Daqing oilfield (Daqing, China). The oil viscosity was 10.3 mPa·s. The ion composition of the formation water is shown in Table 1.

### 2.2. Synthesis of Allyl-OP-10

The synthesis route of Allyl-OP-10 is illustrated in Figure 1. An amount of 20 g of OP-10 and 1.48 g of sodium hydroxide were put into 250 mL of a single neck flask. After stirring for 30 min at 50 °C, 4.5 g of bromo-propylene was added slowly to the mixture and kept on reaction for 24 h. After the reaction, 50 mL of ethanol was poured into the solution and the product (Allyl-OP-10) was obtained using a rotary evaporator at 60 °C.

### 2.3. Synthesis of LMWAP

The synthesis route of LMWAP is illustrated in Figure 2. An amount of 1.2 g of Allyl-OP-10, 2.5 g of AA, 10 g of AM, and 0.11 g of sodium formate were dissolved in water. The pH value of the solution was adjusted to 6–8. Then, the initiator V50 was added dropwise into the previous solution. Finally, the system was placed in a water bath (50 °C) to react for 8 h. The product (LMWAP) was purified and separated using ethanol.

### 2.4. Structure Characterization

The chemical groups of Allyl-OP-10 and LMWAP were analyzed with a Fourier transform infrared (FTIR) spectrophotometer (Bruker, Ettlingen, Germany). The samples were mixed with KBr powder and then pressed into pellets for testing. In the ^1^H-NMR test, samples were dissolved in deuteroxide and then analyzed by nuclear magnetic resonance spectroscopy (Bruker, Germany) to determine the chemical structure. In addition, the morphology of LMWAP was characterized with a scanning electron microscope (SEM, FEI, Hillsboro, OR, USA). Static and dynamic light scattering (SLS, DLS) measurements were performed on a wide-angle light scattering detector (Brookhaven, Holtsville, NY, USA) to study the MW and hydrodynamic radius of LMWAP. All the LMWAP solutions were clarified with a 0.8 μm Millipore filter, and the brines were also clarified with a 0.1 μm Millipore filter to remove dust.

### 2.5. Solution Properties

Polymers were dissolved in fresh water or formation water to prepare polymer solutions with designed concentration and salinity. Then, the viscosity of the polymer solutions was measured by using a DV-2T viscometer (Brookfield, MA, USA). The rheological curves were conducted by using a rheometer (MCR 302, Anton Paar, Tokyo, Japan). In rheological experiments, the shear rate was 0.1 s^−1^ to 1000 s^−1^ and the frequency was 0.1 Hz to 20 Hz.

### 2.6. Interfacial Activity Measurements

The IFTs between the crude oil and different polymer solutions were measured using the spinning drop method with a JJ2000B2 spinning drop IFT apparatus (Zhongchen, Shanghai, China). The polymer solution was primarily filled in a glass tube, and then a droplet of oil was injected into the center of the water phase. Finally, the IFT was measured at a set rotating velocity (6000 rpm) and a given temperature (45 °C).

### 2.7. Emulsification Measurements

Generally, 10 mL of oil and 10 mL of polymer solution were placed in a glass measuring cylinder. The mixture was emulsified using a mechanical stirrer at 500 r/min for 30 min. The emulsion volume was recorded and the emulsion stability was determined by monitoring the emulsion as a function of time. The microstructure of the microgels and emulsion droplets was observed using a DM4500P optical microscope (Leica, Wetzlar, Germany).

### 2.8. Core Flooding Experiments

Five artificial cores with a diameter of 2.5 cm and a length of 10 cm were used in the core flooding experiments. The basic parameters of the cores are listed in Table 2. The radius of the pore throat (*R*_c_) was calculated using the Kozeny–Carmen equation [34,35]:*R*_c_ = [*K*(1 − *φ*)^2^/*φ*·C]^0.5^(1)
where *K* is the water permeability of the core, μm^2^; *φ* is the porosity, %; and C is the Kozeny constant, which is generally 0.2.

The polymer injection experiments were carried out according to the schematic diagram shown in Figure 3. Polymer solutions were injected into the brine-saturated cores separately at 45 °C until the pressure drop reached stable. Subsequent water flooding was then conducted until the pressure drop reached stable again. The flow rate was fixed at 0.5 mL/min. The resistance factor (*RF*) and residual resistance factor (*RRF*) could be calculated according to the following formulas:*RF* = Δ*P*_p_/Δ*P*_wb_(2)
*RRF* = Δ*P*_wa_/Δ*P*_wb_(3)
where Δ*P*_wb_, Δ*P*_p_, and Δ*P*_wa_ refer to the stable injection pressure during water flooding, polymer flooding, and subsequent water flooding, respectively.

In the oil displacement experiments, the cores were first evacuated and saturated with the formation brine. Afterward, they were saturated with crude oil until the water production was zero. Then, water flooding was carried out until the water cut reached 98%. Subsequently, after injecting a 0.3 PV polymer slug, the subsequent water flooding was conducted until the water cut reached 98% again. The flow rate was fixed at 0.1 mL/min. The pressure, water cut, and oil recovery were recorded during the whole oil displacement process.

## 3. Results and Discussion

### 3.1. Characterizations of Allyl-OP-10 and LMWAP

Figure 1 shows the FTIR spectra of OP-10, Allyl-OP-10, and LMWAP. From the curve of OP-10, the bands observed at 3470 cm^−1^ and 2930–2820 cm^−1^ were attributed to the stretching vibration peaks of -OH and -CH_2_-, respectively. The peak area of -OH at 3470 cm^−1^ in the spectrum of Allyl-OP-10 obviously decreased, demonstrating that -OH had been transformed. The peaks of -CH_2_- and benzene were observed in the spectra of LMWAP, demonstrating the introduction of Allyl-OP-10 in the polymer chain. In addition, as illustrated in Figure 2, the H^1^-NMR results further confirmed the structure of OP-10, Allyl-OP-10, and LMWAP. The Zimm plot of LMWAP is shown in Figure 3. By extrapolating to infinite dilution and taking the intercept, the weight-average MW was determined to be 3.9 × 10^6^ mol/g. Furthermore, the intermolecular hydrophobic association contributes to strengthening the interaction between polymer chains and forming a spatial network; therefore, as shown in Figure 4, the morphology of LMWAP exhibited a typical three-dimensional network structure.

### 3.2. Thickening Capability of LMWAP

The thickening capability of a polymer is the primary consideration for achieving mobility control during polymer flooding. The thickening capability of LMWAP was studied in both fresh water and formation water through the relationship between apparent viscosity (*η_s_*) and polymer concentration. As shown in Figure 5a, the viscosity of all the polymers increases with the polymer concentration, and the three polymers all exhibit classical power law behavior. It is well known that the viscometric properties of polymer solutions closely relate to the structure and conformation of polymers in solution. The power law index, a representation of the entangled characteristics of linear polymers [36], is 2.25 for LMWAP in fresh water, which is greater than that of the counterparts HPAM-1800 and HPAM-800, suggesting that LMWAP has the highest thickening efficiency among all the polymers. However, due to its low MW, the viscosity of LMWAP was lower than that of HPAM-1800 at the same concentration. In contrast, as shown in Figure 5b, the viscosity of LMWAP was higher than that of HPAM-1800 when the polymer concentration was higher than 1000 mg/L. The metal cation in formation water significantly weakened the thickening capability of HPAM-1800 and HPAM-800 through charge screening. Meanwhile, the hydrophobic association promoted the formation of intermolecular aggregates, which significantly improved the viscosity retention of LMWAP in formation water. The critical association concentration (CAC) of LMWAP is approximately 870 mg/L and 825 mg/L, respectively, in fresh water and formation water, which demonstrates that the hydrophobic association of LMWAP was enhanced and the intermolecular aggregates were formed at a lower concentration. Therefore, LMWAP exhibited better thickening capability than HPAM-1800 and HPAM-800 in formation water when the polymer concentration was higher than 1000 mg/L. It was reported that the introduction of a long alkyl chain or aromatic ring contributes to improving the thickening ability of water-soluble polymers, especially above the CAC [37]. The carbon number of the alkyl chain or aromatic ring in one hydrophobic monomer has a significant effect on the CAC and thickening ability of polymers. For example, Jiang et al. [38] synthesized a novel double-tailed hydrophobically associating polymer whose hydrophobic monomer possesses two alkyl chains with a carbon number of 22, which is higher than that of LMWAP. The CAC of the reported polymer was only 750 mg/L. In addition, the hydrophobic group content on a polymer also has an effect on the thickening ability. Yang et al. [39] reported that higher hydrophobic group content contributes to improving the association function between hydrophobic groups and the thickening ability of polymers. However, since the carbon number of the hydrophobic group was only 11, the CAC was 1000 mg/L for the sample with the highest hydrophobic group content in the study. Therefore, the increase in the carbon number and content of the hydrophobic monomer could be the method to further improve the thickening ability of LMWAP.

### 3.3. Rheology and Viscoelasticity of LMWAP

Steady rheological experiments are performed to investigate the rheological behavior of the LMWAP solution. Figure 6a shows that the viscosities of all polymer solutions decreased gradually with the increase in shear rate, demonstrating that all the polymer solutions are typical shear-thinning fluids. It is accepted that the physical entanglements and intermolecular forces, such as hydrogen bonds, Van Edward forces, and hydrophobic associations, were gradually broken down as the shear rate increased, resulting in the destruction of the three-dimensional network of polymer molecules and the decrease in viscosity. Due to the molecular structure characteristics, the networks of HPAM-1800 and HPAM-800 were mainly formed by physical entanglements, while the network of LMWAP was formed by both physical entanglements and hydrophobic association. As a result, LMWAP exhibited more significant shear-thinning behavior than HPAM-800 and HPAM-1800, which contributes to improving the injectivity of LMWAP in the near wellbore zone.

The viscoelasticity of the polymer solution relates closely to the micro-displacement efficiency since the polymer with strong elasticity contributed to mobilizing the residual oil at dead ends in reservoirs. The plots of G′/G″ versus the frequency of polymer solutions are presented in Figure 6b–d. The phenomenon occurs in HPAM-1800 and HPAM-800 solutions wherein G′ increases and overtakes G″ along with the frequency measured, which indicates that the solution behaves as a viscous fluid at low frequencies but is elastic at high frequencies. The solution of LMWAP exhibits a similar viscoelastic property but the intersection point of G′ and G″ shifts to a lower frequency. This result indicates that LMWAP exhibited stronger elasticity than HPAM-1800 and HPAM-800 regardless of the fact that the two counterparts possess higher MW. The intermolecular hydrophobic association contributes greatly to the elasticity of LMWAP molecular networks.

### 3.4. IFT Measurements

Oil–water IFT is an important indicator for enhancing oil recovery since the reduction in IFT can significantly reduce the capillary force during the displacement. The decrease in capillary force leads to an increase in the capillary number, which contributes to the mobilization of the remaining crude oil and enhances ultimate oil recovery. The equilibrium IFTs between oil and the LMWAP solution as a function of polymer concentration are shown in Figure 7. The presence of Allyl-OP-10 in molecular chains endows LMWAP with amphiphilicity. As the LMWAP concentrations were increased, the IFTs reduced sharply for the fresh water solution and the formation water solution with 100–3000 mg/L LMWAP, and then decreased slowly above 3000 mg/L LMWAP, which could be attributed to the saturated adsorption of LMWAP molecules at the oil–water interface. Meanwhile, the IFTs reached 2.91 mN/m and 0.88 mN/m, respectively, in fresh water and formation water. LMWAP exhibited better interfacial activity in the formation water, which may be because the electrostatic repulsion generated from -COO^−^ was weakened by cations in formation water and LMWAP formed a tighter adsorption layer at the oil–water interface. Furthermore, the relatively high content of Allyl-OP-10 in LWMAP could be the main reason for its favorable interfacial activity. Normally, the mass ratio of hydrophobic monomer in a polymer used for EOR is less than 3% to ensure water solubility. Therefore, despite the introduction of a hydrophobic monomer such as alkylphenol ethoxylates [40] and (E)-N-(docos-13-enoyl)-N-methylglycine [41], many amphiphilic polymers used for EOR can only reduce IFT to ~10 mN/m. However, the Allyl-OP-10 in LWMAP was almost 8.8%, and the water solubility was guaranteed by its low MW. Likewise, Babu et al. [42] reported a polymeric surfactant whose hydrophobic monomer mass ratio was higher than 50%. The high hydrophobic monomer content endowed the polymeric surfactant with favorable activity, and the IFT could be reduced to 10^−2^ mN/m. Therefore, increasing the content of Allyl-OP-10 while ensuring water solubility is one way to enhance the interfacial activity of LMWAP.

### 3.5. Emulsifying Measurements

In situ emulsification is considered an important micro behavior during chemical flooding because the Jamin effect of emulsion droplets is positive to improve the swept efficiency in micro heterogeneous distributions. As mentioned in Section 3.4, LMWAP possesses an amphiphilic structure and favorable interfacial activity. Therefore, the emulsifying ability of LMWAP was studied at different concentrations in formation water. Figure 8a illustrates the emulsion volume of oil and water at different LMWAP concentrations. The emulsion volume increased from 14 mL to 20 mL as the LMWAP concentration increased from 1000 mg/L to 4000 mg/L, which results from more adsorbed LMWAP molecules at the oil–water interface at higher concentrations. Meanwhile, as shown in Figure 8b, the increased LMWAP concentration also leads to a decrease in the size of emulsion droplets. The average droplet size (D50) decreased from 41.5 um to 8.5 um when the LMWAP concentration increased from 1000 mg/L to 4000 mg/L. According to the Stokes equation, the emulsion systems with smaller droplets possess better stability. In addition, the higher solution viscosity results in better emulsion stability. Therefore, as illustrated in Figure 8c, the emulsions with higher concentrations of LMWAP showed better stability and higher residual emulsion volume after 10 days of aging.

### 3.6. Flow Behavior of Polymers in Porous Media

The flow behavior of polymers reflects the compatibility of the polymer molecular coils and the pore throats, which is an indicator of injectivity and the migration characteristic of polymers. The relationship between pressure drop change and the cumulative injected pore volume (PV) during water flooding, polymer flooding, and subsequent water flooding is illustrated in Figure 9. The pressure drop of LMWAP and HPAM-800 increased progressively with polymer injection before finally reaching an equilibrium stage, implying their smooth transportation in porous media. In contrast, the pressure drop consistently increased as HPAM-1800 was injected into the core, which indicates that it is difficult for HPAM-1800 to migrate in porous media, and strong plugging happened during HPAM-1800 flooding. Considering that the viscosity of LMWAP was higher than HPAM-1800 in formation water, this phenomenon is most likely caused by the large coil size of HPAM-1800, which leads to fatal pore throat plugging. As shown in Table 3, the hydrodynamic sizes of HPAM-1800, HPAM-800, and LMWAP were 186 nm, 124 nm, and 192 nm, respectively, in formation water. The radius of the pore throat of the cores was around 770 nm. It is believed that stable plugging would occur when the ratio of the size of the pore throat to that of the size of the injected polymer is less than 5.0 [34]. The *R*_c_/*R*_h_ value of HPAM-800 is 6.19, which is larger than the threshold (5.0). In contrast, the *R*_c_/*R*_h_ values of LMWAP and HPAM-1800 are 4.13 and 4.05, respectively, which are smaller than the threshold (5.0). However, a plugging issue only happened for HPAM-1800, which may be related to the shear-reversible network structure of LMWAP. As shown in Figure 4, the reversible intermolecular hydrophobic association endowed the network of LMWAP with better flexibility to adapt the space of the pore throats. The physical crosslinked polymer network of LMWAP could be disassociated into separated polymer chains, and this dynamic transition would decrease the size of polymer coils. As a result, the LMWAP molecules passed the narrow throats and re-assembled at the pores of the rock. This repeated micro process enables LMWAP to maintain good injectivity under low-permeability conditions. In contrast, the coils of HPAM cannot undergo dissociation under external forces, which leads to plugging once the size of HPAM exceeds the threshold value. Therefore, LMWAP exhibited favorable injectivity under low-permeability conditions. Furthermore, despite the lower MW, LMWAP exhibited higher *RF* and *RRF* than HPAM-800, which indicates that the capacity of LMWAP to reduce the water–oil mobility ratio and permeability of porous media to water is greater than that of HPAM-800 under identical conditions, which contributes to resulting in higher sweep efficiency.

### 3.7. Enhanced Oil Recovery by LMWAP

The oil displacement experiments were carried out to intuitively reveal the ability of LMWAP for EOR. The concentration of LMWAP used in the oil displacement experiments was set at 1200 mg/L. The concentration of HPAM-800 was set at 1500 mg/L for comparative studies considering the close viscosity. HPAM-1800 was not investigated in this section due to its poor injectivity, as mentioned above. As shown in Figure 10, the pressure gradually increases with the injection of polymer for both LMWAP and HPAM-800. A higher peak pressure (1.88 MPa) for 1200 mg/L LMWAP is taken compared to 1.16 MPa for 1500 mg/L HPAM-800, even though they have the same initial shear viscosities. The stronger sensitivity of the entanglement structure to the elongational stress of the pore throat is considered an important reason for this [33]. In addition, considering the amphiphilicity of LMWAP, in situ emulsification could be the other reason resulting in the difference in pressure. Higher peak pressure usually reflects better mobility control ability, which results in higher sweep efficiency for LMWAP. In addition, as mentioned in Section 3.4, LMWAP could significantly reduce the IFT, which contributes to improving capillary number. Capillary number, understood as the ratio of viscous force to capillary force, is one of the most important parameters in enhanced oil recovery [43]. A higher capillary number leads to a higher micro oil displacement efficiency. The oil recovery at different stages is summarized in Table 4. The better mobility control ability, along with the IFT-reducing ability, endowed LMWAP with significantly higher enhanced oil recovery compared to HPAM-800. A total of 0.3 PV of polymer slug and a subsequent water slug can further increase oil recovery up to 21.5% for LMWAP compared with only 11.5% for HPAM-800. Since the thickening and interfacial activity of LMWAP could be further improved by optimizing the content and alkyl chain length of hydrophobic groups, the structure–activity relationships could be systematically studied in future studies to provide more systematic theoretical guidance for the structural design of the polymers used for EOR in low-permeability reservoirs.

## 4. Conclusions

In summary, an amphiphilic polymer LMWAP with a low MW (3.9 × 10^6^ g/mol) was proposed in this study as a polymer flooding agent for low-permeability reservoirs. Due to the intermolecular hydrophobic association caused by the hydrophobic groups on the polymer chains, LMWAP exhibited better thickening capability in brine than its counterparts, a high-WM HPAM-1800 and a medium-WM HPAM-800. The intermolecular hydrophobic association also endowed LMWAP with more significant shear-thinning behavior and stronger elasticity compared to the two counterparts. Furthermore, despite the higher hydrodynamic size, LMWAP exhibited better injectivity than HPAM-1800 under low-permeability conditions, which relates closely to the reversible association–disassociation transition characteristics of LMWAP. The amphiphilic monomer endowed LMWAP with favorable IFT reduction and emulsification ability. As a result, LMWAP could enhance oil recovery up to 21.5%, which is almost twice as high as that for its counterpart HPAM-800, regardless of the close solution viscosities. The findings of this study provide some enlightenment on designing the structure of polymers used for oil displacement in low-permeability reservoirs. The structure–activity relationships of LMWAP could be future research to build a more systematic theoretical guidance for the structural design of the polymers.

## Data Availability

Data are contained within the article.

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
