# Peer review of "Enhancing Oil Recovery in Low-Permeability Reservoirs Using a Low-Molecular Weight Amphiphilic Polymer"

_polymers, 2024, doi:10.3390/polym16081036_

Round 1

Reviewer 1 Report

Comments and Suggestions for Authors

Reviewer comments

The article entitled, ‘Enhancing Oil Recovery in Low Permeability Reservoirs by a Low Molecular Weight Amphiphilic Polymer’ discusses the role of low molecular weight amphiphilic polymer in oil recovery. The manuscript has been designed well with suitable analytical methods. There are few comments that could improve the quality of the paper, which are given below,

l  The interpretation of FT-IR in figure 1 is not proper. Discuss it elaborately. Baseline correction should also be done.

l  For the compound allyl-OP-10, the -CH2- peak around 2900 – 2800 cm-1 is not clear. Reason it out.

l  Similarly, for the compound LMWAP, the NH2 peak around 3400 cm-1 looks very broad. This broad peak is usually observed for -OH and NH2 peaks shows a sharp peak around this region. Proper explanation should be given in this regard.

l  In case of NMR spectra, there is no explanation with respect to the peaks obtained in the text part. The spectrum in 2b is not clear, there are many impurities found. With respect to the structure, ‘a’ protons in both the compounds show a singlet in fig. 2a, whereas, triplet is seen in fig. 2b. It will be better to purify the compound before taking the analysis.

l  Apart from 1H-NMR, 13C-NMR is also important to predict the structure formation. Include the 13C-NMR for both the compounds, with their discussions in the results part.

Author Response

The article entitled, ‘Enhancing Oil Recovery in Low Permeability Reservoirs by a Low Molecular Weight Amphiphilic Polymer’ discusses the role of low molecular weight amphiphilic polymer in oil recovery. The manuscript has been designed well with suitable analytical methods. There are few comments that could improve the quality of the paper, which are given below,

  1. a) The interpretation of FT-IR in figure 1 is not proper. Discuss it elaborately. Baseline correction should also be done.

Response: The FT-IR spectra have been retested and the analysis of the results of FT-IR has been revised in the revised manuscript.

Modification: Figure 1, Page 10, line 5, “Figure 1 showed the FTIR spectra of OP-10, Allyl-OP-10 and LMWAP. From the curve of OP-10, the bands observed at 3470 cm-1, and 2930-2820 cm-1 were attributed to the stretching vibration peaks of -OH and -CH2-, respectively. The peak area of -OH at 3470 cm-1 in the spectrum of Allyl-OP-10 decreased obviously, demonstrating the -OH has been transformed. The peaks of -CH2- and benzene were observed in the spectra of LMWAP, demonstrating the introduction of Allyl-OP-10 in the polymer chain.” was added.

  1. b) For the compound allyl-OP-10, the -CH2- peak around 2900-2800 cm-1 is not clear. Reason it out.

Response: The FTIR spectra have been retested and the analysis of the results of FTIR has been revised in the revised manuscript.

Modification: Figure 1, Page 10, line 5, “Figure 1 showed the FTIR spectra of OP-10, Allyl-OP-10 and LMWAP. From the curve of OP-10, the bands observed at 3470 cm-1, and 2930-2820 cm-1 were attributed to the stretching vibration peaks of -OH and -CH2-, respectively. The peak area of -OH at 3470 cm-1 in the spectrum of Allyl-OP-10 decreased obviously, demonstrating the -OH has been transformed. The peaks of -CH2- and benzene were observed in the spectra of LMWAP, demonstrating the introduction of Allyl-OP-10 in the polymer chain.” was added.

  1. c) Similarly, for the compound LMWAP, the NH2 peak around 3400 cm-1 looks very broad. This broad peak is usually observed for -OH and NH2 peaks shows a sharp peak around this region. Proper explanation should be given in this regard.

Response: Ethanol was used to purify the polymer. However, in the drying process, the surface of the polymer tended to form compact and strong film to restrain the diffusion of H2O and ethanol within the polymer Therefore, broad peak around 3400 cm-1 was observed, which can be assigned to -OH. The analysis of the results of FTIR has been revised in the revised manuscript.

Modification: Figure 1, Page 10, line 5, “Figure 1 showed the FTIR spectra of OP-10, Allyl-OP-10 and LMWAP. From the curve of OP-10, the bands observed at 3470 cm-1, and 2930-2820 cm-1 were attributed to the stretching vibration peaks of -OH and -CH2-, respectively. The peak area of -OH at 3470 cm-1 in the spectrum of Allyl-OP-10 decreased obviously, demonstrating the -OH has been transformed. The peaks of -CH2- and benzene were observed in the spectra of LMWAP, demonstrating the introduction of Allyl-OP-10 in the polymer chain.” was added.

  1. d) In case of NMR spectra, there is no explanation with respect to the peaks obtained in the text part. The spectrum in 2b is not clear, there are many impurities found. With respect to the structure, ‘a’ protons in both the compounds show a singlet in fig. 2a, whereas, triplet is seen in fig. 2b. It will be better to purify the compound before taking the analysis.

Response: LMWAP has been further purified and then 1H-NMR of LMWAP has been retested. As shown in Figure 1, the chemical shifts are assigned to the target hydrogen in the spectrum. The picture quality of the spectrum is also modified. The triplet is assigned to the hydrogen coming from initiator residue and Allyl-OP-10. Because of the high molecular weight and few dosages of Allyl-OP-10 and initiator, the result of the chemical shift at 1.20 ppm is difficult to further confirm.

Figure 1 1H-NMR spectra of OP-10, Allyl-OP-10 and LMWAP

Modification: Figure 2, Page 10, line 11, “the H1-NMR results further confirmed the structure of OP-10, Allyl-OP-10 and LMWAP.” was added.

  1. e) Apart from 1H-NMR, 13C-NMR is also important to predict the structure formation. Include the 13C-NMR for both the compounds, with their discussions in the results part.

Response: 13C-NMR spectra of OP-10 and allyl-OP-10 are tested and the chemical shifts are assigned to the target carbon in the spectra (Figure 2). Few impurity peaks can be found in 13C-NMR spectra, demonstrating the purity of the product. Because of the high molecular weight, 13C-NMR spectrum of LMWAP is difficult to test. Therefore, few literatures report the 13C-NMR spectrum of such type of water-soluble polymers.

Figure 2 13C-NMR spectra of OP-10 and Allyl-OP-10

Reviewer 2 Report

Comments and Suggestions for Authors

Some suggestions:

XPS are suggested to characterize the LMWAP to explore the chemical constitution and bonding.

The LMWAP cannot be clearly observed in SEM images, so the TEM images are recommended to add for supplementary explanation.

The quality of Figure 2 should be higher for the readers.

Which spindle did you use for viscometer?

What was the density of your sample?

Discussion of the results with other studies was poor.

Author Response

Comments and Suggestions for Authors

Some suggestions:

a) XPS are suggested to characterize the LMWAP to explore the chemical constitution and bonding.

Response: XPS is commonly used for qualitative, quantitative, or semi quantitative, and valence state analysis of the elemental composition on the surface of solid samples. The polymer chains of LMWAP are contracted when the sample exists in a solid state. Therefore, the characterization of chemical bonds on the surface of the solid sample is not the focus for a polymer used for EOR. FT-IR and H1-NMR are the most commonly used method to characterize the polymers used for EOR and XPS characterization was rarely used in relevant studies.[1-4]

[1] Zhou, M.; Yi, R.; Gu, Y.; Tu, H., Synthesis and evaluation of a Tetra-copolymer for oil displacement. Journal of Petroleum Science and Engineering 2019, 179, 669-674.

[2] Chen, H.; Tang, H.; Wu, X.; Liu, Y.; Bai, J.; Zhao, F., Synthesis, Characterization, and Property Evaluation of a Hydrophobically Modified Polyacrylamide as Enhanced Oil Recovery Chemical. Journal of Dispersion Science and Technology 2016, 37, (4), 486-495.

[3] Wu, R.; Zhang, S.; Chen, Y.; Chen, H.; Wang, M.; Tan, Y., Salt endurable and shear resistant polymer systems based on dynamically reversible acyl hydrazone bond. Journal of Molecular Liquids 2022, 346, 117083.

[4] Xiong, P.; Niu, Y.; Meng, F.; Ma, Q.; Song, C.; Zhang, Q.; Che, Y., Grafted copolymers based on HPMC with excellent thermo-viscosifying and salt-tolerant properties at low concentration for enhanced oil recovery. Geoenergy Science and Engineering 2024, 233, 212566.

b) The LMWAP cannot be clearly observed in SEM images, so the TEM images are recommended to add for supplementary explanation.

Response: The introduction of amphiphilic monomer endowed LMWAP with intermolecular hydrophobic association, which contributes to the formation of 3-D network structure. The interaction (hydrophobic association) between polymer chains leads to the change in the micro morphology of the polymer, which could be observed at the micrometer scale. Therefore, many studies regarding polymers for EOR only use SEM for the micro morphology characterization[5-9]. TEM is more suitable for observing the structural changes in nanometer scale and usually uses in the characterization of nanomaterial/polymer composite for EOR.

[5] Wang, R.; Pu, W.; Dang, S.; Jiang, F.; Zhao, S., Synthesis and characterization of a graft-modified copolymer for enhanced oil recovery. Journal of Petroleum Science and Engineering 2020, 184, 106473.

[6] Qin, X. P.; Zheng, J. P.; Li, L. C.; Li, C. X.; Sun, G. L.; Lu, H. W.; Peng, T., Preparation and Performance of an Adsorption Type Gel Plugging Agent as Enhanced Oil Recovery Chemical. JOURNAL OF CHEMISTRY 2015, 2015.

[7] Kang, W.; Zhu, Z.; Yang, H.; Tian, S.; Wang, P.; Zhang, X.; Lashari, Z. A., Study on the association behavior of a hydrophobically modified polyacrylamide in aqueous solution based on host-guest inclusion. JOURNAL OF MOLECULAR LIQUIDS 2019,, 544-553.

[8] Liu, J.; Feng, J.; Yang, S.; Gang, H.; Mu, B., The recovery of viscosity of HPAM solution in presence of high concentration sulfide ions. Journal of Petroleum Science and Engineering 2020, 195, 107605.

[9] Jiang, F.; Feng, X.; Hu, R.; Pang, S.; Pu, W., Synthesis of a novel double-tailed hydrophobically associating polymer for ultra-high salinity resistance. Journal of Molecular Liquids 2022, 367, 120470.

c) The quality of Figure 2 should be higher for the readers.

Response: The quality of Figure 2 has been improved in the revised manuscript.

Modification: Figure 2.

d) Which spindle did you use for viscometer?

Response: The 18# rotor (5.6r/min, 7.34s-1) was used for viscometer, which was set according to the standard of petroleum industry in China.

e) What was the density of your sample?

Response: The density of the sample is approximately 1.3 g/cm3. In fact, the polymers used for EOR are all water-soluble products. The polymers have negligible effect on the density of polymer solutions due to the low used dosage and the density of polymer solutions was quite close to the water or brine used for preparing polymer solutions.

f) Discussion of the results with other studies was poor.

Response: The results of other studies were comparatively discussed in the revised manuscript.

Modification: Page 13, line 15, “It was reported that the introduction of long alkyl chain or aromatic ring contributes to improve the thickening ability of water-soluble polymers, especially above the CAC[37]. The carbon number of the alkyl chain or aromatic ring in one hydrophobic monomer has significant effect of the CAC and thickening ability of polymers. For example, Jiang et al.[38] synthesized a novel double-tailed hydrophobically associating polymer whose hydrophobic monomer possesses two alkyl chains and the carbon number was 22, which is higher than that of LMWAP. The CAC of the reported polymer was only 750 mg/L. In addition, the hydrophobic group content on a polymer also has an effect on the thickening ability. Yang et al.[39] reported that higher hydrophobic group content contributes to improve the association function between hydrophobic groups and the thickening ability of polymers. However, since the carbon number of the hydrophobic group was only 11, the CAC was 1000 mg/L for the sample with the highest hydrophobic group content in the study. Therefore, the increase in the carbon number and content of the hydrophobic monomer could be the method to further improve the thickening ability of LMWAP.” was added. Page 17, line 1, “Furthermore, the relatively high content of Allyl-OP-10 in LWMAP could be the main reason for its favorable interfacial activity. Normally, the mass ratio of hydrophobic monomer in a polymer used for EOR is less than 3% to ensure the water-solubility. Therefore, despite the introduction of hydrophobic monomer such as alkylphenol ethoxylates[40] and (E)-N-(docos-13-enoyl)-N-methylglycine[41], many amphiphilic polymers used for EOR can only reduced IFT to ~10 mN/m. However, the Allyl-OP-10 in LWMAP was almost 8.8% and the water-solubility was guaranteed by its low MW. Likewise, Babu et al.[42] reported a polymeric surfactant whose hydrophobic monomer mass ratio was higher than 50%. The high hydrophobic monomer content endowed the polymeric surfactant with favorable activity and the IFT could be reduced to 10-2 mN/m. Therefore, increasing the content of Allyl-OP-10 while ensuring water solubility is one way to enhance the interfacial activity of LMWAP.” was added.

Reviewer 3 Report

Comments and Suggestions for Authors

The paper discusses the challenges and advancements in polymer flooding for enhanced oil recovery (EOR) in low-permeability reservoirs. It highlights the importance of EOR methods due to the limitations of conventional energy sources and the significant oil reserves in low-permeability reservoirs in China.

The study emphasizes the critical role of selecting the right polymer for effective polymer flooding. The paper explores the synthesis and performance of an amphiphilic polymer with a low-MW, demonstrating improved thickening capability, shear-thinning behavior, and oil-water interfacial activity compared to traditional HPAM variants.

Various functionalized polymers have been developed to address injectivity issues in low-permeability reservoirs. Encapsulated polymers and self-adaptive polymers show promise in enhancing injectability and oil recovery rates, offering new pathways for polymer design in EOR applications.

The study provides a detailed experimental methodology for synthesizing polymers, characterizing their structures, measuring solution properties, assessing interfacial activity, conducting emulsification tests, and performing core flooding experiments. These experiments offer valuable insights into the performance of LMWAP in low-permeability conditions.

The research sheds light on the molecular design of polymers for EOR in low-permeability reservoirs, offering potential solutions to the challenges faced in polymer flooding. The findings open avenues for further research in optimizing polymer properties, enhancing injectivity, and improving oil recovery rates in challenging reservoir conditions.

The detailed experimental approach, coupled with the synthesis and performance evaluation of LMWAP, provides a solid foundation for future research endeavors aimed at enhancing the efficiency and effectiveness of polymer flooding in challenging reservoir environments.

The following aspects may be improved:

·       Detailed Analysis of Polymer Synthesis: The review can delve deeper into the synthesis process of LMWAP, discussing the specific reactions, conditions, and mechanisms involved in creating this polymer.

·       A more extensive comparison with other polymers commonly used in EOR can provide a clearer understanding of the advantages and limitations of LMWAP.

·       Providing a detailed analysis of how LMWAP enhances injectivity under low permeability conditions and its effectiveness in improving oil recovery rates compared to traditional polymers can be beneficial.

·       A thorough discussion on the core flooding experiments conducted, the methodology employed, and the results obtained can strengthen the review's experimental insights.

·       Exploring the molecular design aspects of LMWAP, such as the role of amphiphilicity, hydrophobic association, and dynamic transition ability, can enrich the discussions.

·       Potential optimizations in polymer design, and areas for further experimentation can enhance the scientific value of the paper.

By expanding on these topics and providing a detailed analysis of each aspect, the paper can improve its scientific value and contribute to the advancement of knowledge in the field of enhanced oil recovery using polymers in low-permeability reservoirs.

Comments on the Quality of English Language

The English is of good quality

Author Response

The paper discusses the challenges and advancements in polymer flooding for enhanced oil recovery (EOR) in low-permeability reservoirs. It highlights the importance of EOR methods due to the limitations of conventional energy sources and the significant oil reserves in low-permeability reservoirs in China.

The study emphasizes the critical role of selecting the right polymer for effective polymer flooding. The paper explores the synthesis and performance of an amphiphilic polymer with a low-MW, demonstrating improved thickening capability, shear-thinning behavior, and oil-water interfacial activity compared to traditional HPAM variants.

Various functionalized polymers have been developed to address injectivity issues in low-permeability reservoirs. Encapsulated polymers and self-adaptive polymers show promise in enhancing injectability and oil recovery rates, offering new pathways for polymer design in EOR applications.

The study provides a detailed experimental methodology for synthesizing polymers, characterizing their structures, measuring solution properties, assessing interfacial activity, conducting emulsification tests, and performing core flooding experiments. These experiments offer valuable insights into the performance of LMWAP in low-permeability conditions.

The research sheds light on the molecular design of polymers for EOR in low-permeability reservoirs, offering potential solutions to the challenges faced in polymer flooding. The findings open avenues for further research in optimizing polymer properties, enhancing injectivity, and improving oil recovery rates in challenging reservoir conditions.

The detailed experimental approach, coupled with the synthesis and performance evaluation of LMWAP, provides a solid foundation for future research endeavors aimed at enhancing the efficiency and effectiveness of polymer flooding in challenging reservoir environments.

The following aspects may be improved:

a) Detailed Analysis of Polymer Synthesis: The review can delve deeper into the synthesis process of LMWAP, discussing the specific reactions, conditions, and mechanisms involved in creating this polymer.

Response: The synthesis of LMWAP was based on free radical polymerization method. The reaction conditions such as temperature, chemical concentration, pH and so on have effect on the molecular weight and thickening ability of the product. This study is a proof-of-concept study which focuses on the benefits of the structure characters including low molecular weight and amphiphilicity on the injectivity and oil recovery ability of polymers used for EOR in low permeability reservoirs. Systematic study on synthesis methods, conditions, and mechanisms will be carried out in the future.

b) A more extensive comparison with other polymers commonly used in EOR can provide a clearer understanding of the advantages and limitations of LMWAP.

Response: HPAM is the most typical and commonly used oil displacement polymer in the petroleum industry. Therefore, this study used medium molecular weight and high molecular weight HPAM as counterparts for comparative study. In the revised manuscript, more study results of polymers were supplemented and discussed with LMWAP.

Modification: Page 13, line 15, “It was reported that the introduction of long alkyl chain or aromatic ring contributes to improve the thickening ability of water-soluble polymers, especially above the CAC[37]. The carbon number of the alkyl chain or aromatic ring in one hydrophobic monomer has significant effect of the CAC and thickening ability of polymers. For example, Jiang et al.[38] synthesized a novel double-tailed hydrophobically associating polymer whose hydrophobic monomer possesses two alkyl chains and the carbon number was 22, which is higher than that of LMWAP. The CAC of the reported polymer was only 750 mg/L. In addition, the hydrophobic group content on a polymer also has an effect on the thickening ability. Yang et al.[39] reported that higher hydrophobic group content contributes to improve the association function between hydrophobic groups and the thickening ability of polymers. However, since the carbon number of the hydrophobic group was only 11, the CAC was 1000 mg/L for the sample with the highest hydrophobic group content in the study. Therefore, the increase in the carbon number and content of the hydrophobic monomer could be the method to further improve the thickening ability of LMWAP.” was added. Page 17, line 1, “Furthermore, the relatively high content of Allyl-OP-10 in LWMAP could be the main reason for its favorable interfacial activity. Normally, the mass ratio of hydrophobic monomer in a polymer used for EOR is less than 3% to ensure the water-solubility. Therefore, despite the introduction of hydrophobic monomer such as alkylphenol ethoxylates[40] and (E)-N-(docos-13-enoyl)-N-methylglycine[41], many amphiphilic polymers used for EOR can only reduced IFT to ~10 mN/m. However, the Allyl-OP-10 in LWMAP was almost 8.8% and the water-solubility was guaranteed by its low MW. Likewise, Babu et al.[42] reported a polymeric surfactant whose hydrophobic monomer mass ratio was higher than 50%. The high hydrophobic monomer content endowed the polymeric surfactant with favorable activity and the IFT could be reduced to 10-2 mN/m. Therefore, increasing the content of Allyl-OP-10 while ensuring water solubility is one way to enhance the interfacial activity of LMWAP.” was added.

c) Providing a detailed analysis of how LMWAP enhances injectivity under low permeability conditions and its effectiveness in improving oil recovery rates compared to traditional polymers can be beneficial.

Response: The superior injectivity and oil displacement ability of LMWAP compared to traditional polymers have been discussed and the possible mechanism has been illustrated.

Modification: Scheme 4 was added. Page 20, line 9, “The physical crosslinked polymer network of LMWAP could be disassociated into separated polymer chains and this dynamic transition decreased the size of polymer coils. As a result, the LMWAP molecules passed the narrow throats and re-assembled at the pores of the rock. This repeated micro process enables LMWAP to maintain good injectivity under low permeability conditions. In contrast, the coils of HPAM cannot undergo dissociation under external forces, which leads to plugging once the size of HPAM exceeds the threshold value. Therefore, LMWAP exhibited favourable injectivity under low-permeability conditions.” was added.

d) A thorough discussion on the core flooding experiments conducted, the methodology employed, and the results obtained can strengthen the review's experimental insights.

Response: The discussion regarding core flooding experiments has been enhanced in the revised manuscript.

Modification: Page 22, line 9, “In addition, as mentioned in Section 3.4, LMWAP could significantly reduced the IFT, which contributes to improve capillary number. Capillary number, understood as the ratio of viscous force to capillary force, is one of the most important parameters in enhanced oil recovery[43]. A higher capillary number leads to a higher micro oil displacement efficiency.” was added.

e) Exploring the molecular design aspects of LMWAP, such as the role of amphiphilicity, hydrophobic association, and dynamic transition ability, can enrich the discussions.

Response: The effects of amphiphilicity, hydrophobic association, and dynamic transition ability on LMWAP have been discussed in the revised manuscript.

Modification: Page 13, line 15, “It was reported that the introduction of long alkyl chain or aromatic ring contributes to improve the thickening ability of water-soluble polymers, especially above the CAC[37]. The carbon number of the alkyl chain or aromatic ring in one hydrophobic monomer has significant effect of the CAC and thickening ability of polymers. For example, Jiang et al.[38] synthesized a novel double-tailed hydrophobically associating polymer whose hydrophobic monomer possesses two alkyl chains and the carbon number was 22, which is higher than that of LMWAP. The CAC of the reported polymer was only 750 mg/L. In addition, the hydrophobic group content on a polymer also has an effect on the thickening ability. Yang et al.[39] reported that higher hydrophobic group content contributes to improve the association function between hydrophobic groups and the thickening ability of polymers. However, since the carbon number of the hydrophobic group was only 11, the CAC was 1000 mg/L for the sample with the highest hydrophobic group content in the study. Therefore, the increase in the carbon number and content of the hydrophobic monomer could be the method to further improve the thickening ability of LMWAP.” was added. Page 17, line 1, “Furthermore, the relatively high content of Allyl-OP-10 in LWMAP could be the main reason for its favorable interfacial activity. Normally, the mass ratio of hydrophobic monomer in a polymer used for EOR is less than 3% to ensure the water-solubility. Therefore, despite the introduction of hydrophobic monomer such as alkylphenol ethoxylates[40] and (E)-N-(docos-13-enoyl)-N-methylglycine[41], many amphiphilic polymers used for EOR can only reduced IFT to ~10 mN/m. However, the Allyl-OP-10 in LWMAP was almost 8.8% and the water-solubility was guaranteed by its low MW. Likewise, Babu et al.[42] reported a polymeric surfactant whose hydrophobic monomer mass ratio was higher than 50%. The high hydrophobic monomer content endowed the polymeric surfactant with favorable activity and the IFT could be reduced to 10-2 mN/m. Therefore, increasing the content of Allyl-OP-10 while ensuring water solubility is one way to enhance the interfacial activity of LMWAP.” was added.

f) Potential optimizations in polymer design, and areas for further experimentation can enhance the scientific value of the paper.

Response: The potential optimizations in polymer structure and further study area regarding polymer flooding in low permeability reservoirs has been discussed in the revised manuscript.

Modification: Page 22, line 17, “Since the thickening and interfacial activity of LMWAP could be further improved by optimizing the content and alkyl chain length of hydrophobic groups, the structure-activity relationships of could be systematically studied in future studies to provide more systematic theoretical guidance for the structural design of the polymers used for EOR in low permeability reservoirs.” was added.

By expanding on these topics and providing a detailed analysis of each aspect, the paper can improve its scientific value and contribute to the advancement of knowledge in the field of enhanced oil recovery using polymers in low-permeability reservoirs.

Reviewer 4 Report

Comments and Suggestions for Authors

This manuscript is interesting for materials and polymers science community and could be accepted for publication. The topic is up to date and actual. Polymer flooding has achieved considerable success in medium-high permeability reservoirs. However, when it comes to low-permeability reservoirs, polymer flooding suffers from poor injectivity due to the large molecular size of the commonly used high-molecular-weight (high-MW) partially hydrolyzed polyacrylamides (HPAM). Herein, an amphiphilic polymer (LMWAP) with a low-MW (3.9 × 106 g/mol) was synthesized by introducing an amphiphilic monomer (Allyl-OP-10) and a chain transfer agent into the polymerization reaction. Despite the low-MW, LMWAP exhibited better thickening capability in brine than its counterparts HPAM-1800 (MW=1.8×107 g/mol) and HPAM-800 (MW=8×106 g/mol) due to the intermolecular hydrophobic association. LMWAP also exhibited more significant shear-thinning behavior and stronger elasticity than the two counterparts. Furthermore, LMWAP possesses favorable oil-water interfacial activity due to its amphiphilicity. The oil-water interfacial tension (IFT) could be reduced to 0.88 mN/m and oil-in-water (O/W) emulsions could be formed under the effect of LMWAP. In addition, the reversible hydrophobic association endows the molecular chains of LMWAP with dynamic association-disassociation transition ability. Therefore, despite the similar hydrodynamic sizes in brine, LMWAP exhibited favorable injectivity under low permeability conditions, while the counterpart HPAM-1800 led to fatal plugging. Furthermore, LMWAP could enhance oil recovery up to 21.5%, while the counterpart HPAM-800 could only enhance oil recovery up to 11.5%, which could be attributed to the favorable interfacial activity of LMWAP. The subject addressed in this article is worthy of investigation. The information presented is new. The methodology of research is appropriate. The conclusions supported by the data. The manuscript is good illustrated and interesting to read. I have only two suggestions for minor revision. First, it would be a good idea to cite some additional relevant references about other perspective compounds in introduction, e.g. Polymer 2021 212, 123119.Second, some more detailed perspectives about the future research could be formulated in conclusions.

Author Response

This manuscript is interesting for materials and polymers science community and could be accepted for publication. The topic is up to date and actual. Polymer flooding has achieved considerable success in medium-high permeability reservoirs. However, when it comes to low-permeability reservoirs, polymer flooding suffers from poor injectivity due to the large molecular size of the commonly used high-molecular-weight (high-MW) partially hydrolyzed polyacrylamides (HPAM). Herein, an amphiphilic polymer (LMWAP) with a low-MW (3.9 × 106 g/mol) was synthesized by introducing an amphiphilic monomer (Allyl-OP-10) and a chain transfer agent into the polymerization reaction. Despite the low-MW, LMWAP exhibited better thickening capability in brine than its counterparts HPAM-1800 (MW=1.8×107 g/mol) and HPAM-800 (MW=8×106 g/mol) due to the intermolecular hydrophobic association. LMWAP also exhibited more significant shear-thinning behavior and stronger elasticity than the two counterparts. Furthermore, LMWAP possesses favorable oil-water interfacial activity due to its amphiphilicity. The oil-water interfacial tension (IFT) could be reduced to 0.88 mN/m and oil-in-water (O/W) emulsions could be formed under the effect of LMWAP. In addition, the reversible hydrophobic association endows the molecular chains of LMWAP with dynamic association-disassociation transition ability. Therefore, despite the similar hydrodynamic sizes in brine, LMWAP exhibited favorable injectivity under low permeability conditions, while the counterpart HPAM-1800 led to fatal plugging. Furthermore, LMWAP could enhance oil recovery up to 21.5%, while the counterpart HPAM-800 could only enhance oil recovery up to 11.5%, which could be attributed to the favorable interfacial activity of LMWAP. The subject addressed in this article is worthy of investigation. The information presented is new. The methodology of research is appropriate. The conclusions supported by the data. The manuscript is good illustrated and interesting to read. I have only two suggestions for minor revision.

a) First, it would be a good idea to cite some additional relevant references about other perspective compounds in introduction, e.g. Polymer 2021 212, 123119.

Response: The reference has been carefully consulted and the authors considered that this reference is not closely related to this study. In future related research, citations of this reference will be considered.

b) Second, some more detailed perspectives about the future research could be formulated in conclusions.

Response: The perspectives about the future research has been supplemented in the revised manuscript.

Modification: Page 24, line 5, “The structure-activity relationships of LMWAP could be future research to build a more systematic theoretical guidance for the structural design of the polymers.” was added.

Round 2

Reviewer 2 Report

Comments and Suggestions for Authors

The authors have well addressed the comments from the reviewer. It is now acceptable.